# Absorbed dose calculation for a realistic CT-derived mouse phantom irradiated with a standard Cs-137 cell irradiator using a Monte Carlo method

**Amir Entezam**[1,2,3]*, **Andrew Fielding**[2,4], **David Bradley**[5,6], **Davide Fontanarosa**[1,2]

**1** School of Clinical Sciences, Queensland University of Technology, Brisbane, QLD, Australia, **2** Centre for Biomedical Technologies, Queensland University of Technology, Brisbane, QLD, Australia, **3** Translational Research Institute, The University of Queensland, Brisbane, QLD, Australia, **4** School of Chemistry and Physics, Faculty of Science, Queensland University of Technology, Brisbane, QLD, Australia, **5** Centre for Applied Physics and Radiation Technologies, Sunway University, PJ, Malaysia, **6** Department of Physics, University of Surrey, Guildford, United Kingdom

* amir.entezam@hdr.qut.edu.au

**Data Availability Statement:** All relevant data are within the manuscript and its Supporting Information files.

## Abstract

Computed tomography (CT) derived Monte Carlo (MC) phantoms allow dose determination within small animal models that is not feasible with in-vivo dosimetry. The aim of this study was to develop a CT-derived MC phantom generated from a mouse with a xenograft tumour that could then be used to calculate both the dose heterogeneity in the tumour volume and out of field scattered dose for pre-clinical small animal irradiation experiments. A BEAMnrc Monte-Carlo model has been built of our irradiation system that comprises a lead collimator with a 1 cm diameter aperture fitted to a Cs-137 gamma irradiator. The MC model of the irradiation system was validated by comparing the calculated dose results with dosimetric film measurement in a polymethyl methacrylate (PMMA) phantom using a 1D gamma-index analysis. Dose distributions in the MC mouse phantom were calculated and visualized on the CT-image data. Dose volume histograms (DVHs) were generated for the tumour and organs at risk (OARs). The effect of the xenographic tumour volume on the scattered out of field dose was also investigated. The defined gamma index analysis criteria were met, indicating that our MC simulation is a valid model for MC mouse phantom dose calculations. MC dose calculations showed a maximum out of field dose to the mouse of 7% of $D_{max}$. Absorbed dose to the tumour varies in the range 60%-100% of $D_{max}$. DVH analysis demonstrated that tumour received an inhomogeneous dose of 12 Gy-20 Gy (for 20 Gy prescribed dose) while out of field doses to all OARs were minimized (1.29 Gy-1.38 Gy). Variation of the xenographic tumour volume exhibited no significant effect on the out of field scattered dose to OARs. The CT derived MC mouse model presented here is a useful tool for tumour dose verifications as well as investigating the doses to normal tissue (in out of field) for pre-clinical radiobiological research.

**Funding:** This work was supported by Queensland University of Technology grant (Institute of Health and Biomedical Innovation (IHBI) John Williams Cancer Research Scholar). he funders had no role in study design, data collection and analysis, decision to publish, or preparation of the manuscript.

**Competing interests:** The authors declare that they have no known competing financial interests or personal relationships that could have appeared to influence the work reported in this paper.

# 1 Introduction

The increased interest in small-animal radiation biology research has resulted in a requirement for improvements in the standardized dosimetry of pre-clinical irradiations [1], [2, 3]. In vivo dose measurement can be accomplished using devices like MOSFET-based dosimeters [4–6], Gafchromic films [7], or silica-based fibres as thermoluminescent dosimeters (TLDs) [8–14]. Computational techniques using mathematical models of systems can also be used to calculate radiation doses. Over the past twenty years the calculation based approach has become increasingly more widespread due to the constantly growing availability and power of computers [15]. One essential application of these computerized methods is to create computational phantoms where geometries and materials can be defined. The methods utilized to create the geometry of computational models in anthropomorphic phantoms have progressed from portraying structures using quadratic equations to producing voxel-based representations [15–19]. Similarly, the advancement in creating computational models of small animals has developed from the mathematical equation based phantom models [16, 20] to subsequently advanced voxel-based models [21–23].

Imaging modalities like micro-CT [24] are used to produce high-resolution three-dimensional images from small animals [25]. CT data can be used to generate voxelated representations of volumes and the associated tissues in computational phantoms. CT data also allows the introduction of corrections for the tissue inhomogeneities into dose calculations during treatment planning procedures [26, 27]. These corrections involve the determination of a relationship between tissue electron density and their corresponding Hounsfield Units (HU) [28]. Accurate dose calculations within computational phantoms have become achievable due to the advancement of MC simulation methods [29]. Accordingly, using CT data to generate phantoms that can simulate anatomically correct small animal models as well as accomplishing accurate MC dose calculations within the phantoms allows providing a standard method for reliable dosimetric analysis in pre-treatment verifications [30]. In fact, this technique benefits from replicating the clinical treatment planning procedures in a pre-clinical setting [31]. In the present study, we used this method for dosimetric verification of xenograft tumour-bearing mouse irradiations.

Small animal-derived tumour xenografts have emerged as valuable preclinical model systems in radiation biology cancer research [32–34]. Tumours are often implanted in small animals subcutaneously and targeted with a conformal radiation field at different stages of the tumours' growth [35]. These xenographic tumours, depending on the position and size, can generate complex dose calculations, different from the standard dosimetry for tumours within the small animal body. One such complexity is tumour dose inhomogeneity which has effects on radiation biology cancer studies [36]. Also, the level of absorbed doses to normal tissue (out of field dose), caused by irradiating the xenograft tumours, can affect the parameters under biological investigation. Therefore, it is necessary to accurately investigate the absorbed dose in the preclinical xenograft tumour models (with different sizes) and in normal tissues when the tumours are irradiated with collimated beams. This is the primary goal of this research. This study is required for our radio-immunotherapy (combination of radiotherapy (RT) with immunotherapy [37]) investigations, with one of the main interests being in the induced systemic immune response distal to the initial irradiation site [38, 39]. It is critical to ensure that the tumour is irradiated with the prescribed dose as well as ensuring that doses to critical organs are low enough to avoid triggering an immune response directly. Several studies have investigated the dosimetric verification of mouse models using mouse CT data previously, but they often have not presented xenograft models [40–44]. Also, these studies have not investigated the effect of xenographic tumour dimensions on the dosimetry. To address this issue, we

generated a MC model from a micro-CT image set of a mouse with a xenograft tumour for absorbed dose investigations. In this study, we use the term 'MC mouse phantom' to refer to the CT-derived MC model of the mouse. The MC mouse phantom was used to calculate the dose distributions and analyze DVHs for the tumour and OARs. Also, to explore the effect of tumour volume variations on the dosimetry, the xenographic tumour diameter was modified and DVHs were reanalysed.

This study also involves physical dose measurement for verification of MC modelling of the irradiation system. Verification of Monte Carlo simulations is commonly accomplished by comparing dose distributions obtained from physical dose measurements with the corresponding calculated doses. Good agreement between the measured and calculated doses guarantees the accuracy of MC modelling. Several studies have used dosimeters within slabs of PMMA phantom or simple water phantoms to measure the doses, comparing them with the MC simulation results [45–50]. In this study, we use the same method for the verification of our MC simulations.

## 2 Materials and methods

### 2.1 MC mouse phantom

A MC mouse phantom was produced from a whole-body micro-CT (Molecubes X-Cube micro-CT, USA) image set (in DICOM format) of a mouse with 1 cm diameter growing xenographic tumour. The CT image set was then converted to an "egsphant" file (EGSnrc-based MC phantom file) using in-house MATLAB scripts (MATLAB version R2017b, MathWorks Inc, Natick, MA, USA). Every voxel in the MC mouse phantom corresponds to a specific type of mouse tissue with its electron density, from the CT numbers in the image set. The materials employed in the MC mouse phantom include AIR521ICRU, LUNG521ICRU, ICRUTIS-SUE521ICRU, ICRPBONE521ICRU [45], representing air, lung, soft tissue, and bone. The CT number limits for tissue segmentation were -1000 to -750 for air, -750 to -250 for lung, -250 to 350 for soft tissue, and 350 to 1000 for bone. The mouse phantom voxel size was set as $(0.2 \times 0.2 \times 0.3)$ mm$^3$ for x-, y-, and z-axis respectively with 400, 400, and 358 voxels in the X, Y, and Z planes respectively. There is a trade-off between voxel size and statistical accuracy for MC calculations in small animal [45]. The selected voxel size in this study allows for sufficient statistical accuracy while providing adequate resolution for small animal phantoms [45, 46]. In our experimental irradiation setup, as detailed in our previous study [49], mice were positioned on a handmade polystyrene support (air equivalent material with zero attenuation in circular shape) for irradiations and a rubber band was used to secure the tumour within the collimated beam and distance the mouse healthy tissue from the high dose exposure of the beam axis. To generate an accurate MC mouse phantom for absorbed dose calculation, the mouse was scanned along with the handmade polystyrene support and the rubber band.

### 2.2 MC simulation of the irradiation system

**2.2.1 Irradiation system.** We have previously constructed an add-on lead collimator to allow for targeted irradiation of mice [49]. In previous studies, we showed that our collimator combined with a 137Cs source irradiator is an effective method for irradiating small animal xenograft tumours [49–51]. The simulation of MC mouse phantom irradiation was performed using a 1-cm diameter beam produced by the collimator [49] mounted onto a Gammacell 40 Exactor irradiator unit (Best Theratronics, ON K2K OE4, Canada) located at Transitional Research Institute (TRI), Brisbane, Australia. The irradiator holds two $^{137}$Cs sources at the top and bottom of the main chamber (with 0.662 MeV photon energy and total nominal activity of 111 TBq), generating 1.0 Gy/min nominal dose rate at the time of installation. The collimator

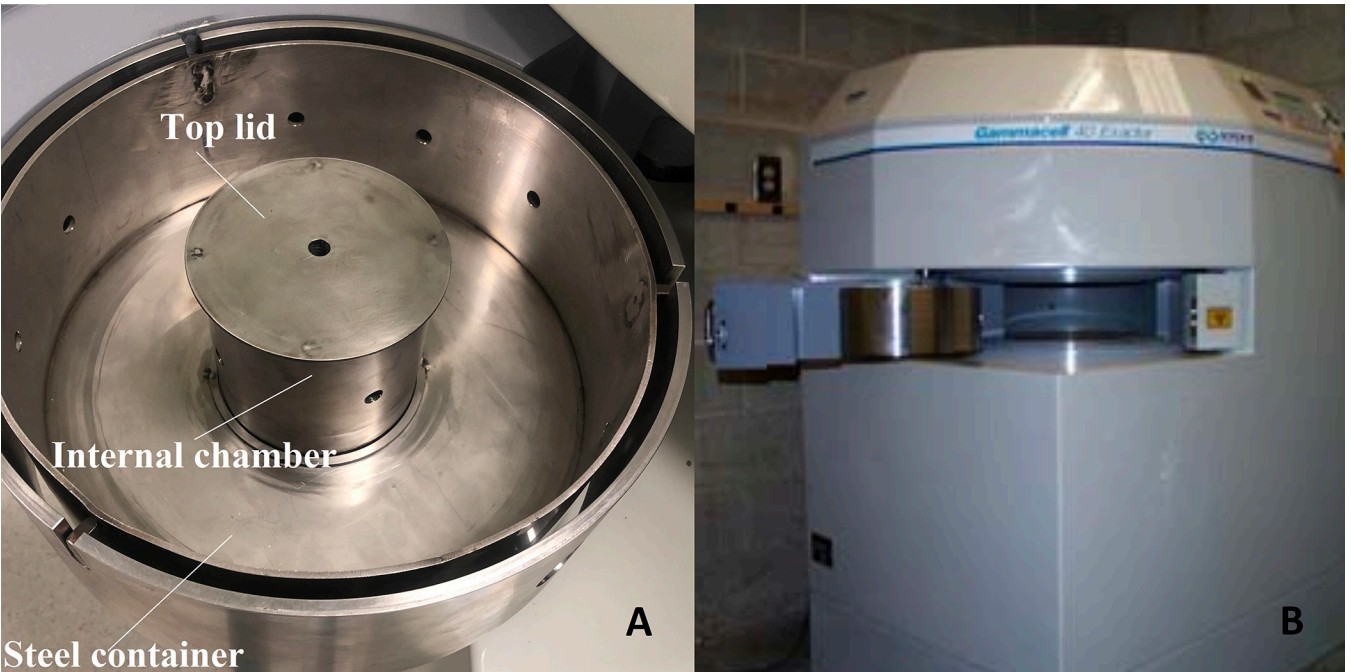

**Fig 1. Add-on collimator and Gammacell irradiator.** A) Add-on collimator components including the top lid, 1cm aperture, internal chamber, and the steel container to hold the collimator inside the irradiator. B) Collimator placed in the Gammacell 40 Exactor, prepared for irradiation.

consists of two 25 mm thick lead lids with 10 mm diameter apertures in the centre, collimating the [137]Cs sources' beams to a 11-mm diameter circular beam (including the penumbra of the beam). The lead collimator is 10 cm in height and diameter, with an internal chamber for placing mouse or phantom for irradiations. A steel container holds the collimator inside the Gammacell. Fig 1 shows all components of the add-on collimator as well as the Gammacell 40 Exactor.

**2.2.2 Phase-space file generation.** The BEAMnrc user code associated with EGSnrc [52, 53] was used to generate the phase-space file. The code was run on a Linux Cluster system with a large university managed central cluster. FLATFILT14 and CONS3r (component modules of BEAMnrc code) were utilized to model the main beam geometry (generated by [137]Cs source positioned 30.5 cm from the add-on collimator's topmost surface) and the geometry of the lead lid with 10 mm aperture. $10^9$ initial gamma-ray photon histories were used to generate the phase-space file, providing information of all particles that cross the scoring-plane. The scoring plane was situated underneath the base surface of the collimator's top lid. Since the collimator's lid has a 10-mm aperture, particles that have reached the scoring plane can mostly pass through the aperture together with a small proportion of photons transition from the lead lid and the scattered radiation. The phase space file was calculated with a single source of the irradiator (the top source) and the top lid of the collimator. Then the phase space file was used at beam angles of 0 and 180 degrees for all dose calculations in order to model the irradiation setup of the dual sources of the irradiator. This allows for the calculation of the total radiation dose (produced by both top and bottom sources) within the targets. Fig 2 shows BEAMnrc component modules and the position of the scoring plane along with the phase space file used multi-directionally at beam angles of 0˚ and 180˚. Electron impact ionization, atomic relaxations, and low energy photon interactions, such as Rayleigh scattering and bound Compton scattering, have been included in our MC simulations [53, 54]. Also, through the condensed

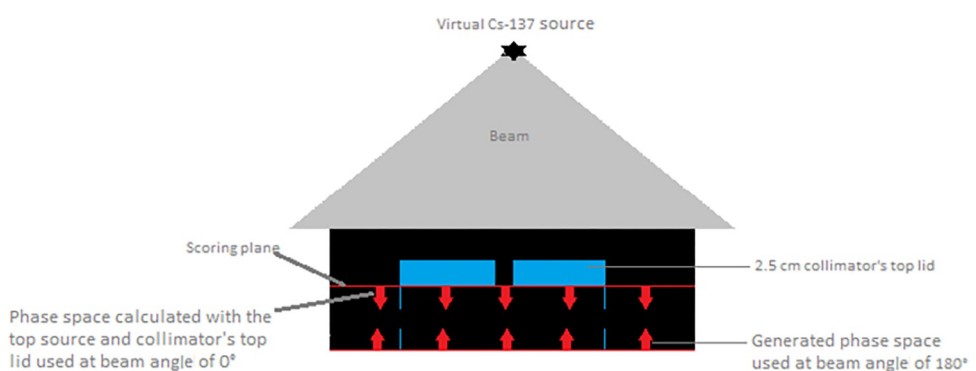

**Fig 2. Illustration of MC modelling of the irradiation system.** MC modelling of the $^{137}$Cs source and collimator top lid are demonstrated using BEAMnrc components. The scoring surface is positioned right below the collimator's lid (red line). The top and bottom red arrows represent the phase-space file used multi-directionally employed at beam angel of 0˚ and 180˚.

history technique, catastrophic inelastic scattering and bremsstrahlung were taken into account in the transport of secondary electrons [54]. Global Electron transport cut-offs (ECUT) and Global photon transport cut-offs (PCUT) were 0.521 MeV and 0.01 MeV respectively [52]. Also, the low energy thresholds for electrons and photons AE and AP were set to 0.521 and 0.01, consistent with the ECUT and PCUT parameters, respectively. These well-established cut-off settings allow for efficient dose calculations while reducing the CPU calculation time [48]. The phase space file was calculated once in this study and was employed for all subsequent dose calculations.

## 2.3 Verification of MC irradiation modelling

To ensure accurate phase-space generation and verify that our MC modelling of the irradiation system is reliable for any specific irradiation condition, we compared measured dose profiles obtained from a cubic polymethyl methacrylate (PMMA) phantom dose measurement (irradiated with our irradiation system) with the corresponding calculated profile obtained from the MC modelling of the PMMA phantom irradiation. The description of measuring and calculating the dose profiles are given in the next two sections.

**2.3.1 PMMA phantom dose measurement.** A PMMA phantom, manufactured by our research group [50], was used for dose measurement. The phantom was made of a PMMA slab with dimensions of $(6.75 \times 2.25 \times 1.50)$ cm$^3$ and an extrusion with diameter of $(0.50 \times 0.50 \times 0.75)$ cm$^3$ was implanted on the side to represent the radiological characteristics and approximate shape of a mouse soft tissue with a xenograft tumour model in the flank as demonstrated in Fig 3A. The phantom was sliced into two equivalent slabs to allow for the secure placement of radiochromic film. A piece of Gafchromic EBT3 film (Ashland Inc, Covington, Kentucky, USA) was first placed on the handmade polystyrene in the collimator and was irradiated. Then the corresponding field was marked on the polystyrene support to indicate the collimated field. A rectangular piece of Gafchromic EBT3 film was sandwiched between the slabs and the phantom was placed inside the collimator, positioning the sandwiched Gafchromic EBT3 film right at the midplane of the collimator. Fig 3A also shows the Gafchromic EBT3 film sandwiched between PMMA slabs. The tumour model was accurately positioned within the indicated collimated beam and the PMMA phantom was irradiated to obtain dose profile measurement. Fig 3B demonstrates the placement of the PMMA phantom in the main chamber of the collimator on the polystyrene, with the tumour positioned within the collimated beam ready for irradiation. Film dosimetry was performed based on the

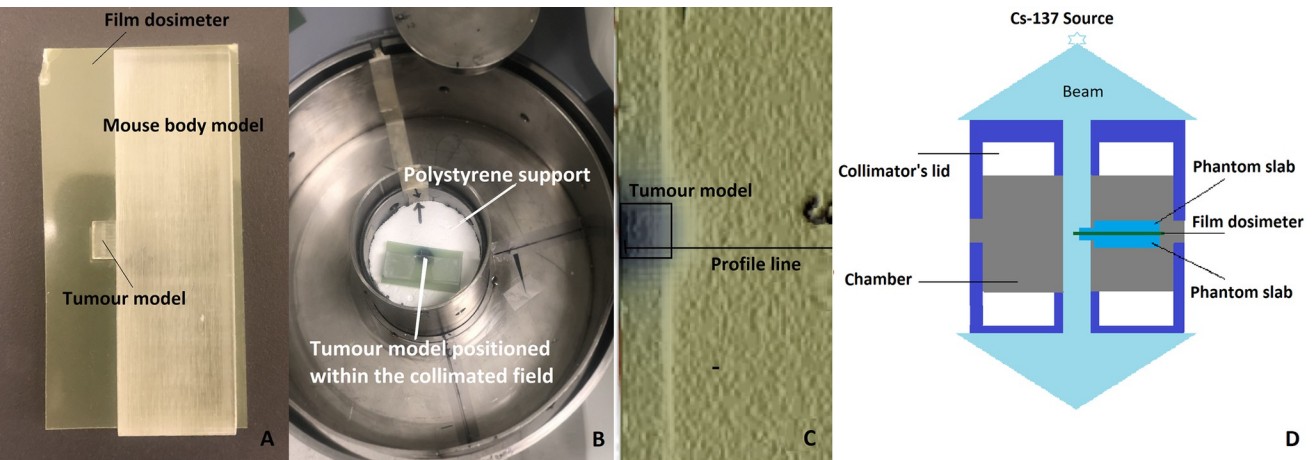

**Fig 3. Illustration of the EBT3 film sandwiched between PMMA slabs and irradiation setup.** A) The GAFchromic film dosimeter sandwiched between PMMA phantom slabs. B) The PMMA phantom was placed inside the collimator and the tumour model was situated within the marked collimated beam on the polystyrene platform for irradiation. C) The irradiated film from the phantom irradiation is shown. Irradiation field and profile line selection are marked. D) Cutaway drawing of the collimator with the PMMA phantom positioned at the midplane of the collimator and the film dosimeter sandwiched between phantom slabs irradiated with Cs-137 beam. Film dosimeter, phantom slabs, and the collimator's components are marked.

Gafchromic film dosimetry procedure given in previous studies [50, 55], and a dose profile measurement was obtained. The irradiated film along with the profile line and the region where the film was exposed is demonstrated in Fig 3C.

**2.3.2 PMMA phantom dose calculation.** MC modelling of the PMMA phantom irradiation was performed and phantom doses were calculated. Material-specific data files for the PMMA phantom were created using EGSnrc PEGS4 [52]. The experimental irradiation set-up of the PMMA phantom (which was described in the previous section) was replicated using DOSXYZnrc (a user code for EGSnrc) [56]. The voxels for the PMMA phantom and outside the phantom were set to PMMA521ICRU and ICRUAIR521 (material name for PMMA and air in EGSnrc). The phase-space source (as described in section 2.2.2) was positioned at 0 and 180 degrees to the phantom geometry the particles transported through the voxelized phantom. The tumour model centre point was set as the isocentre for phase-space file and dose deposited in each voxel was calculated and written to the '.3DDOSE' file (default output file of DOSXYZnrc code). The interactions explained in section 2.2.2 were also included in DOSXYZnrc simulations [56]. $5 \times 10^9$ gamma-ray photon histories were used for dose calculation [47].

**2.3.3 Comparison of the measured and calculated dose profiles.** Gamma index analysis was used to provide a quantitative comparison between the calculated and measured dose profiles. Gamma analysis is the most effective method for comparing two dose distributions [57]. It is established as the gold standard in verification procedures among all available methods and clinical decisions are made based on its outcomes [57, 58]. Gamma analysis allows for comparing two dose distributions with respect to dose and the space domain. In this method, distance to agreement (DTA) (distance between a reference dose point and the closest evaluated dose point) as well as dose difference ($\Delta$D) metrics are merged into a unitless quantity through the following equation, known as the Gamma value [58].

$$\gamma(\overrightarrow{r}_{ref}, \overrightarrow{r}_m) = \min\left\{\sqrt{\frac{|\overrightarrow{r}_{ref} - \overrightarrow{r}_m|^2}{DTA^2} + \frac{|D(\overrightarrow{r}_{ref}) - D(\overrightarrow{r}_m)|^2}{\Delta D^2}}\right\} \qquad (1)$$

where

$|\mathrm{D}(\overrightarrow{r}_{ref}) - \mathrm{D}(\overrightarrow{r}_m)|$ is the dose difference and $|\overrightarrow{r}_{ref} - \overrightarrow{r}_m|$ is the distance between the dose points.

In this way, an acceptance region (an ellipse) is created around every single point of the reference dose distribution. This ellipse should compass the evaluated dose point to pass the gamma test. Mathematically, gamma values of less than 1 imply that the gamma test was passed [57]. Gamma values of less than 1 imply that the gamma test was passed [58]. The gamma analysis was implemented using in-house MATLAB scripts. The EGSnrc dose calculation was set as the reference dose distribution and 2 mm and 2% acceptance limits were utilized for DTA and ΔD, respectively [58].

## 2.4 MC mouse phantom dose calculation

The dose distribution on the MC mouse phantom was calculated using the DOSXYZnrc code. The MC mouse phantom (egsphant file) was used as an input file in the DOSXYZnrc. The coordinate of the tumour centre point was found via DICOM Viewer software [59] in the CT image set, using real-time coordinate viewer tool, and was set as the isocentre for the phase space file (that was generated and validated in sections 2.2.2 and 2.3 respectively). The distance between the scoring plane and the isocentre was set to 5 cm, positioning the MC mouse phantom at the midplane of the collimator MC model, where the mice are positioned for irradiations in our experimental irradiation setup. The phase-space file and number of the gamma-ray photon histories in DOSXYZnrc were set following the setup explained in section 2.3.2 for dose calculation and the 3DDOSE file was obtained.

## 2.5 Dose distribution visualization and DVHs analysis

Evaluation of the MC mouse phantom calculation requires an analysis of the calculated dose distribution, through visual dose review and dose-volume histograms. The Computational Environment for Radiotherapy (CERR) [60], a MATLAB based toolkit, was used for dose visualization and volume contouring for DVH analysis. To visualize the calculated doses on the image set, both the CT-image set and 3DDOSE files were loaded into CERR and coronal and sagittal isodose lines and dose colourwash were displayed. The tumour, heart, liver, Llung (left lung), Rlung (right lung), Lkidney (left kidney), Rkidney (right kidney), and spinal column were contoured on each slice of the CT-image set for DVHs generation. To perform an accurate contouring for the tumour and all OARs, a whole-body mouse ATLAS from coregistered x-ray CT data of a normal mouse was used as guideline [61]. To convert the calculated MC doses to absolute dose for DVH analysis, the calibration method proposed by Popescu et al [62] was employed. Then the DVHs were plotted for 20 Gy absolute prescribed dose.

## 2.6 Xenographic tumour size modification

The xenographic tumours were divided into small, medium, and large groups with diameters of 0.5 cm, 0.75 cm, and 1 cm, respectively, based on irradiation studies on a cohort of mice in various preclinical radiobiological research. Thus, it is vital to determine the effect that xenograft tumour size may have on the MC mouse dosimetry. 3Dslicer software version 4.10.2 (http://www.slicer.org) was used to modify the dimension of the tumour model in the original CT data. The mouse was scanned when the xenographic tumour diameter was 1 cm, which was the maximum tumour dimension observed in all our radio-immunotherapy investigations [49]. To generate new image sets with smaller tumours, the tumour diameter in the initial CT-image set was reduced to 0.75 cm and 0.5 cm, and changes were saved as two new CT image sets. The reduction of the tumour diameter was achieved by setting the voxels in the perimeter region of the tumour to a HU of air using 'segment editor' and 'simple filter' tools in 3Dslicer.

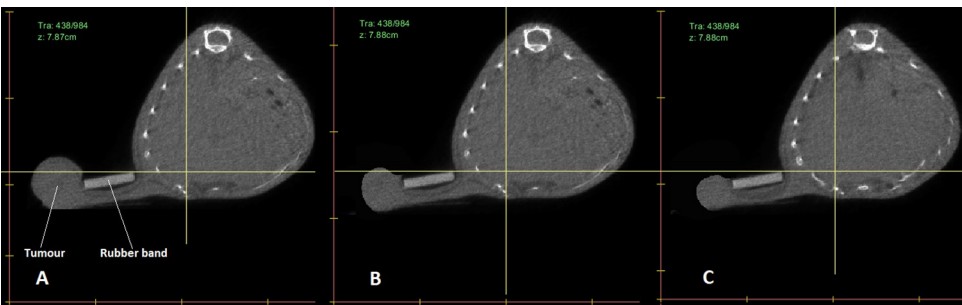

**Fig 4. Mouse CT image set.** Image A demonstrates the transversal view of a slice of the original mouse CT image set with the maximum xenographic tumour diameter of 1 cm. Images B and C demonstrate the modified tumours with diameters of 0.75 cm, and 0.5 cm, respectively. The xenographic tumour and the rubber band that distances the tumour from the mouse healthy tissue are marked in image A.

This allows for producing two new CT image sets and then generating the corresponding MC mouse phantoms with a similar mouse structure and smaller tumour diameters (as demonstrated in Fig 4) for reanalysing the DVHs in order to explore the effect of tumour volume variations on the dosimetry in and out of the main radiation field.

## 3 Results

### 3.1 Verification of MC irradiation modelling

To verify the MC simulation of the irradiation system, the profile from the PMMA phantom experimental dose measurement was compared with the corresponding calculated PMMA phantom EGSnrc profile as shown in Fig 5A. The MC calculated doses for all points along with the corresponding measured doses, corresponding to the profiles shown in Fig 5A, are presented in Table 1. Also, the errors for all MC calculated doses were obtained from the 3ddose file and presented in the last column of the table. Both dose profiles were normalized to their respective maximum doses. The gamma index between the Gafchromic EBT3 film measurement and EGSnrc calculation is demonstrated in Fig 5B. The gamma test was passed by all gamma index values down to 2.25 cm from the beam axis, confirming that both penumbra regions and the out of field doses (at 0.8 cm-2.25 cm from the beam axis) are in an excellent agreement for EBT3 film measurements and EGSnrc calculation. Only two gamma values were smaller than 1 and therefore failed the gamma test. The major deviations between the dose profiles might arise from the statistical nature of the MC simulation as well as uncertainties in the GAFchromic film dosimetry as detailed in the discussion section.

### 3.2 MC mouse phantom dose calculations

The calculated dose distribution within the mouse phantom was visualized on the CT image set via isodose lines and dose colourwash in CERR (Figs 6 and 7). All doses were normalized to $D_{max}$ (maximum dose). As shown in Fig 6, the tumour is on the left side of the mouse body and the rubber band maintains the mouse body distance from the beam. The tumour has been exposed to 60%-100% of $D_{max}$ while only a maximum of 7% scattered dose has been observed within the rest of the mouse body. Doses in areas closer to the centre of the tumour are between 90% and 100% of $D_{max}$ whereas the doses to the edges of the tumour is a maximum 60% of $D_{max}$. As in Fig 7, the green, yellow, and orange isodose lines indicate that absorbed dose in the surface area of the tumour varies in the range 60%-75% of $D_{max}$. As it can be seen in Fig 7, the purple isodose line, which encircles the mouse body structure, indicates that maximum 7% of $D_{max}$ has been distributed within the mouse body.

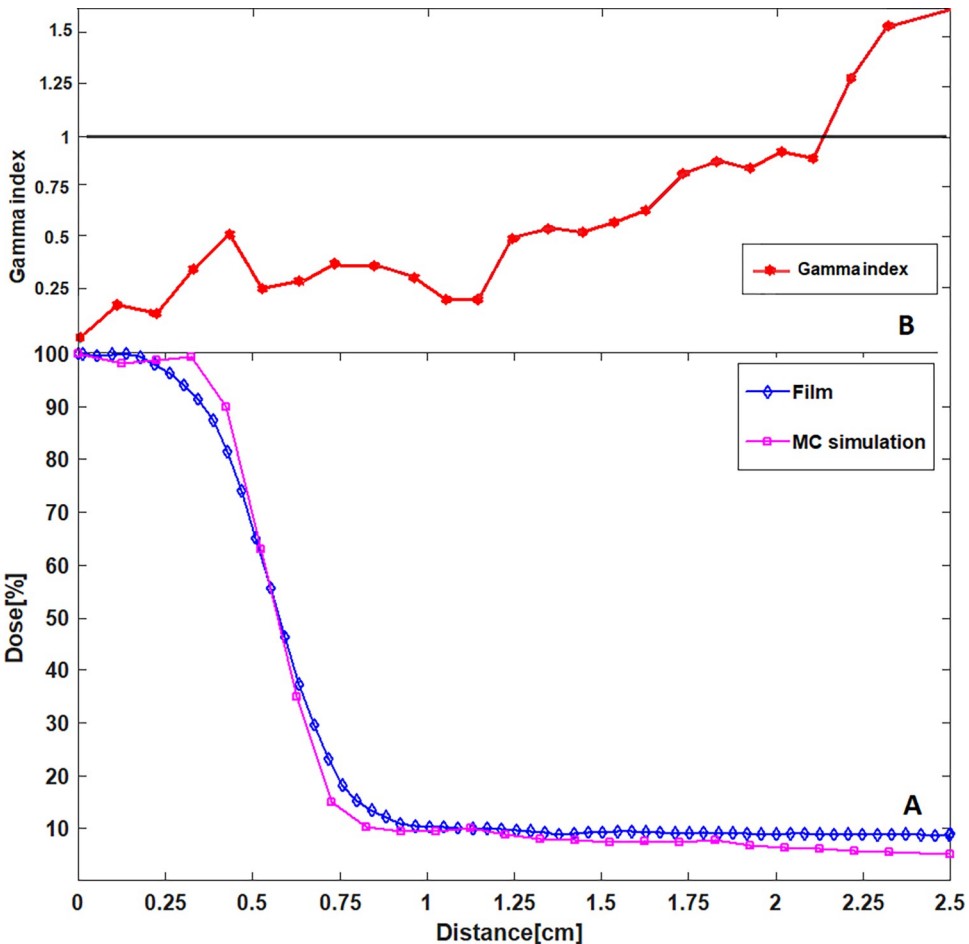

**Fig 5. Measured and calculated dose profiles.** A) The blue line is the dose profile sampled measured across PMMA tumour phantom exposure. The purple line is the corresponding calculated dose profile results from the EGSnrc simulation, B) The cyan line shows the resultant gamma index trend determined between the two relative GAFchromic film measurements and EGSnrc calculation dose profiles.

### 3.3 DVH calculations

Dose calculation of the MC mouse phantom was evaluated by generating and comparing absolute DVHs for tumour and OARs. The DVHs of the tumour, heart, liver, Llung, Rlung, Lkidney, Rkidney, and spinal column are shown for tumour diameters of 1 cm, 0.75 cm, and 0.5 cm in Fig 8. The minimum, maximum, and mean doses for tumour and critical organs were obtained from CERR and are summarized in Table 2 for all tumour irradiations. The DVH indicates that the tumour volume dose varies between 12 Gy-20 Gy. As it can be seen in Fig 8, there is good agreement between the tumours DVHs for the three different sized tumours. This indicates that approximately an equal level of the prescribed dose was delivered to entire volume of three tumours. However, further analysis indicated that as the tumour diameter increased, the mean dose of the tumour slightly increased, and therefore DVH curve slightly shifted to the right as demonstrated in Fig 8. This can also be seen in the tumours' mean dose results (Table 2) where the mean doses for tumour diameters of 0.5 cm, 0.75 cm, and 1 cm are 16.75 Gy, 16.92 Gy, and 17.22 Gy, respectively.

Due to the localisation and immobilisation of the tumour during irradiation, mouse body was effectively spared and therefore absorbed doses to all OARs were minimized (1.29 Gy-1.38

**Table 1. Measured and calculated doses and MC dose calculation errors.**

| Distance (cm) | Measured doses (%) | MC Calculated doses (%) | MC calculation errors (%) |
|---|---|---|---|
| 0 | 100 | 100 | 0.88 |
| 0.12 | 99.44 | 97.53 | 0.79 |
| 0.22 | 97.62 | 98.13 | 0.65 |
| 0.32 | 94.05 | 99.75 | 1.18 |
| 0.42 | 82.37 | 90.42 | 0.78 |
| 0.52 | 62.30 | 62.64 | 0.85 |
| 0.62 | 34.74 | 35.31 | 1.62 |
| 0.72 | 15.73 | 14.32 | 1.92 |
| 0.82 | 13.02 | 11.85 | 2.10 |
| 0.92 | 10.57 | 10.61 | 2.92 |
| 1.02 | 10.45 | 11.12 | 2.79 |
| 1.12 | 11.20 | 11.48 | 3.29 |
| 1.22 | 9.61 | 9.82 | 2.91 |
| 1.32 | 9.60 | 9.01 | 3.22 |
| 1.42 | 9.66 | 8.61 | 2.99 |
| 1.52 | 9.09 | 8.73 | 2.82 |
| 1.62 | 9.60 | 8.67 | 3.30 |
| 1.72 | 9.50 | 8.94 | 1.98 |
| 1.82 | 9.38 | 9.02 | 3.21 |
| 1.92 | 9.43 | 8.92 | 2.96 |
| 2.02 | 9.46 | 7.39 | 3.34 |
| 2.12 | 9.54 | 7.40 | 2.92 |
| 2.22 | 9.41 | 7.13 | 3.07 |
| 2.32 | 9.32 | 6.82 | 2.82 |
| 2.50 | 9.39 | 6.60 | 2.98 |

Dose values for all points of the calculated profile along with the corresponding measured doses (normalized to their respective maximum doses). Dose calculation errors are presented in the last column.

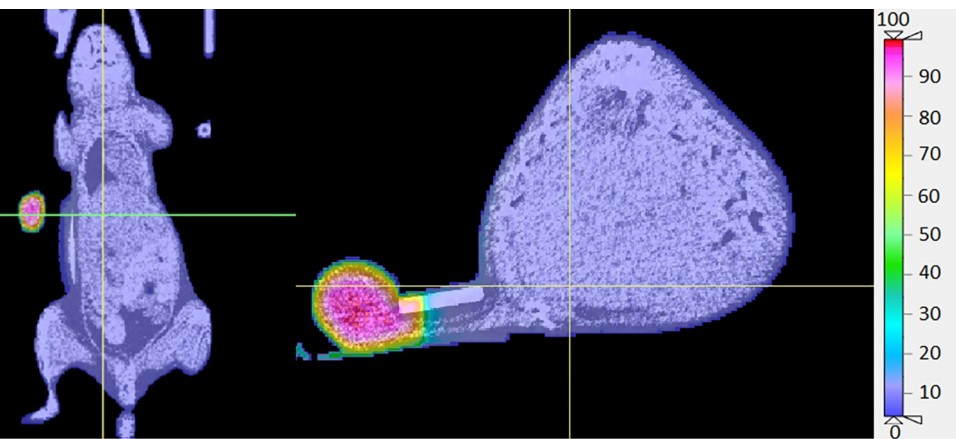

**Fig 6. Dose distribution colourwash demonstration on micro-CT images in CERR.** Doses are normalized to $D_{max}$. The rubber band squeezes the mouse body out of the field and the accurate irradiation modelling of the tumour is demonstrated. The dose colourwash shows approximately 60%-100% of $D_{max}$ to the whole tumour and maximum 7% of $D_{max}$ to the normal tissue.

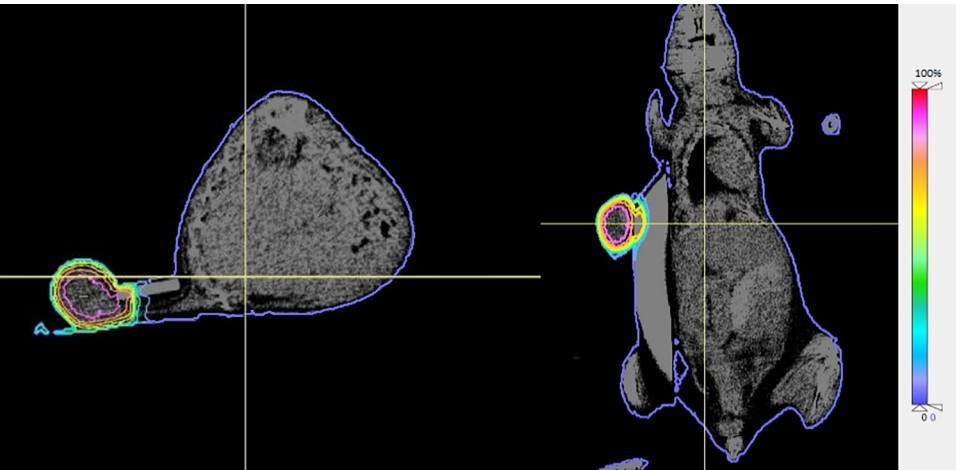

**Fig 7. Isodose demonstration on micro-CT images in CERR.** The sagittal (left) and coronal (right) isodose lines are demonstrated. Doses are normalized to the $D_{max}$. Isodose lines within the tumour show an increase in dose level from the edge to the centre of the tumour. The purple isodose encircling the mouse body indicates the constant dose level of 7% of $D_{max}$ to the mouse body.

Gy) as shown in Table 2. This leads to maximum 0.14 Gy mean dose difference within the out of field region across different organs for three tumour irradiations. This can be seen in isodose line visualizations of cross-sectional slices through the phantom too (Fig 7). This causes sharp DVHs for all organs in each plot. The DVHs of OARs are in a good agreement in the plot due to very good sparing of the OARs as well as dose homogeneity across different organs (Fig 8).

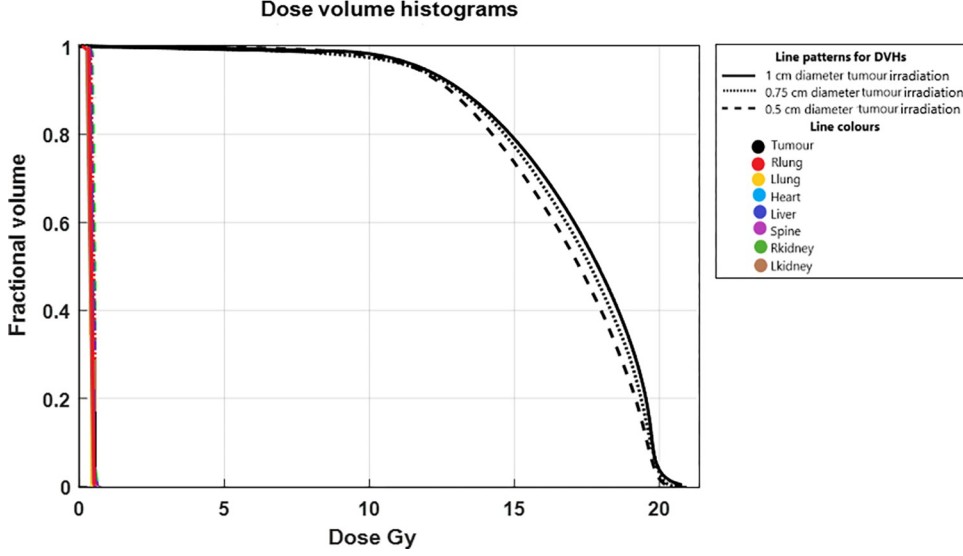

**Fig 8. DVHs plots.** DVHs of the tumours and OARs for 0.5 cm, 0.75 cm, and 1 cm tumour irradiations are demonstrated in dash, dot, and solid lines, respectively. DVHs of the tumours are in black and colours of the DVHs for OARs are presented in the graph.

**Table 2. Minimum, maximum, and mean doses for the tumour and OARs.**

| OARs | Minimum dose (Gy) | | | Maximum dose (Gy) | | | Mean dose (Gy) | | |
|---|---|---|---|---|---|---|---|---|---|
| | Tumour diameter (cm) | | | Tumour diameter (cm) | | | Tumour diameter (cm) | | |
| | 0.5 | 0.75 | 1 | 0.5 | 0.75 | 1 | 0.5 | 0.75 | 1 |
| Rlung | 0.73 | 0.72 | 0.75 | 1.35 | 1.38 | 1.36 | 1.10 | 1.20 | 1.21 |
| Llung | 0.84 | 0.85 | 0.79 | 1.36 | 1.37 | 1.33 | 1.20 | 1.21 | 1.13 |
| Spine | 0.92 | 0.95 | 0.94 | 1.37 | 1.36 | 1.29 | 1.14 | 1.24 | 1.18 |
| Heart | 0.90 | 0.95 | 0.88 | 1.35 | 1.32 | 1.34 | 1.22 | 1.23 | 1.20 |
| Liver | 0.94 | 0.87 | 0.95 | 1.34 | 1.33 | 1.38 | 1.21 | 1.20 | 1.22 |
| Lkidney | 0.92 | 0.85 | 0.95 | 1.33 | 1.29 | 1.32 | 1.19 | 1.16 | 1.21 |
| Rkidney | 0.89 | 0.85 | 0.95 | 1.36 | 1.31 | 1.37 | 1.20 | 1.18 | 1.22 |
| Tumour | 12.22 | 12.05 | 12.57 | 20.67 | 20.53 | 20.73 | 16.75 | 16.92 | 17.22 |

Minimum, maximum, and mean doses (in Gy) for the xenographic tumour with 0.5 cm, 0.75 cm, and 1 cm diameters and critical organs in out of field.

## 4 Discussion

Dose distribution determination and DVH analysis are required for preclinical radiobiological research, for tumour dose verifications as well as investigating the absorbed doses to normal tissue (in out of field) in small animal models. It is critical to ensure that the tumour is treated with the desired dose and that the absorbed dose to normal tissue (in out of field) has minimal impact on the objectives of the study. Also, it is important to verify that the out of field dose will not harm critical organs such as the lungs or brain. Dose calculations and DVHs analysis results justify the use of our MC phantom for small animal irradiations as they provide information on the level of tumour dose and scattered dose surrounding the target prior to live animal RT.

Dose analysis performed in this work leads to several observations. One of the first is the good agreement between the EGSnrc and Gafchromic film dose profile plots, as demonstrated in Fig 5, verifying that our MC modelling of the irradiation system has sufficient accuracy for MC mouse phantom dose calculations. As mentioned in the introduction section, several studies have used the same method for verifying MC simulations. This provides strong support for the validity of the verification method used in this study, i.e., comparing the doses measured within a PMMA phantom with the corresponding calculated MC doses. It is worth mentioning that we used PMMA material in this study because it is tissue equivalent, has a reproducible composition, is easy to machine, and it is commonly used for radiation dosimetry investigations [63–67]. Also, PMMA phantoms are typically employed for standard dosimetry practices such as calibration of in-phantom dose rate [68, 69]. PMMA is found to be an excellent soft-tissue substitute as its material properties such as effective atomic number and electronic density are close to that of soft tissue [70]. So, the in-house built PMMA phantom well mimics soft tissue composition of the mouse. Similar to our method, in research conducted by Kuess et al [71] the mouse tissues were considered PMMA equivalent for dosimetric verification purposes.

The quality indicator for comparison of the profiles is based on the gamma analysis method, with gamma values smaller than 1 indicative of good agreement between the GAFchromic film measurement and EGSnrc calculation dose profiles. The gamma test was passed by 93 percent of the gamma values. Only two gamma index values in out of field dose region did not pass the gamma test. This may be due in part to the statistical nature of the MC simulation, with fewer photons crossing further away from the collimated field resulting in fewer scored particles in MC calculations. This increases the uncertainty in out of field dose

calculations. A mean uncertainty of 3% was obtained in out-of-field dose calculations. To reduce the uncertainty, especially in low-dose areas, it would be necessary to increase the number of photon histories, significantly increasing also CPU calculation times. Additionally, ±2% errors in homogeneity throughout a sheet of Gafchromic EBT3 film [72] and curvature of EBT3 film pieces during the scanning procedure [73] can be other sources of errors. To ensure the accuracy of our Gafchromic EBT3 film measurement, the measured doses were compared against the corresponding doses obtained from MOSKIN dosimeter as detailed in our previous study [49]. Despite all sources of uncertainties, our MC calculation and experimental measurement were found to be well matched, with a mean discrepancy of 4%. With this high level of agreement, we can confirm the accuracy of our MC modelling for MC mouse phantom dose calculation.

Calculation of dose distribution within the MC mouse phantom showed a maximum out of field dose of 7% of $D_{max}$ on the MC mouse phantom body. Both dose wash and isodose lines demonstrate that 7% of $D_{max}$ has been uniformly distributed to all tissues in the MC mouse phantom within the out of field region. This suggests that the phantom heterogeneity almost did not affect the dose within the out of field region across different organs. This can be seen in mean dose results to the OARs in Table 2 too as the maximum dose difference between different organs is only 0.14 Gy. As it is demonstrated in Fig 6, the collimated beam is well aligned with the target and covers the entire tumour as the rubber band restrains the mouse healthy tissue from the high dose exposure of the beam. This suggests that the rubber band is a useful tool to secure the tumour in the radiation field as well as sparing the mouse body from the irradiation when xenographic tumour irradiation is intended in preclinical studies. The isodose lines demonstrated that the rubber band and polystyrene support do not cause any extra scattering to the mouse body. Isodose lines also indicated that absorbed dose to entire tumour varies in the range 60%-100% of $D_{max}$. This is due to the electron disequilibrium which is the result of inadequate build-up region in the edges of the tumours. In fact, a certain depth of tumour tissue is required before the radiation reaches its maximum amount. So, this causes dose escalation in the centre of the tumour and leads to the tumour dose heterogeneity.

DVH analysis results suggest that irradiation of mice with our irradiation method can deliver an inhomogeneous dose to the xenographic tumour as well as minimizing the absorbed dose to other critical organs. The tumour DVHs indicate that it is quite challenging to deliver a homogeneous prescribed dose to the whole xenographic tumour volume. DVHs were plotted for 20 Gy, which is the prescribed dose in all our radio-immunotherapy investigations. However, as it can be seen in Fig 8, only 10% of 1 cm diameter tumour volume is exposed to 20 Gy. Due to the tumour dose heterogeneity (approximately 12 Gy-20 Gy), the mean dose to the tumour is 17.22 Gy (Table 2). To achieve a homogenous tumour dose, the tumour could be covered with a water equivalent bolus in order to provide adequate build-up for beam delivery to the tumour. The other method to deliver the prescribed dose to the whole tumour volume is by delivering a significantly higher dose to the tumour centre to achieve clinically acceptable prescribed tumour dose coverage (e.g., 95% of the prescription dose) everywhere. It must be highlighted that our specific study ultimately aims at investigating abscopal effects of RT [50], for which we do not need to deliver a homogeneous dose to the whole target [49], but only deliver enough dose to a certain fraction of the target [74, 75]. So, at this stage, it was not essential to employ a bolus for tumour irradiations in our experiments. Future directions of this study will aim to achieve and investigate homogeneous radiation dose to tumour using the method we offered in this study.

The DVHs of organs show that the maximum out of field dose to OARs is 1.38 Gy when tumours are irradiated with 20 Gy prescribed dose. This is a desirable out of field dose in our radio-immunotherapy investigations as such a low dose does not induce the immune response directly [76, 77] therefore allowing us to investigate any true induced systemic immune response stimulated by irradiating the xenographic tumour. It is also unlikely that 1.38 Gy out-

of-field dose causes any radiation harm to OARs [78]. The good agreement between three DVH plots for small, medium, and large tumours in Fig 8 implies that variation of the xenographic tumour volume has no significant effect on the DVH of OARs out of the collimated field. This indicates that the scattered radiation caused by increasing the tumour diameter is insignificant. Based on this, the dosimetric verifications performed in the experiments are valid at all stages of mice irradiations, when tumours grow within the range of the investigated tumour dimensions.

There are important advantages of our findings on dose distributions in mouse micro-CT data. Firstly, the issues arising in small animal dosimetry for the heterogeneous material properties of murine subjects are more complex than many research groups have addressed in their studies by using simplified approaches like water equivalent phantoms or simplified mouse phantoms [3, 79]. Alternatively, our approach allows for the demonstration of the applicability of MC phantoms to provide accurate dose data of actual murine subjects applying accurate material properties. Secondly, most studies that applied mouse CT data to perform dosimetric verification of mice irradiations have employed image-guided sophisticated small animal RT systems. So, there is still a lack of a workflow for dosimetric evaluation of small animal irradiations using irradiation methods that have no imaging capabilities like our system. Our study offers a useful method of dosimetric evaluation of small animal RT irradiated with irradiation systems with no imaging capabilities. This provides a quantitative method for characterizing such small animal irradiation methods.

## 5 Conclusion

The CT-derived MC model of a mouse with a xenographic tumour we have introduced is a useful tool for radiation dosimetry investigations within small animal models in various preclinical radiobiological research. The gamma test was passed by 93 percent of the gamma values, confirming the validity of our MC modelling for MC mouse phantom dose calculation. Dose calculation and DVHs results demonstrated that the tumour has been exposed to 60%-100% of Dmax (corresponding to 12 Gy-20 Gy for a 20 Gy prescription dose) while the scattered dose within the rest of the mouse body was only a maximum of 7%. The tumours DVHs also showed that the tumours have been exposed to heterogeneous doses, suggesting that it is challenging to deliver a homogeneous prescribed dose to the whole xenographic tumour volume. The DVHs of OARs showed a maximum out of field dose of 1.38 Gy across different organs. This is low enough to not directly cause the immune response, allowing us to explore the induced systemic immune response distal to the initial irradiation site. Variation of the xenographic tumour volume exhibited no significant effect on the DVHs of OARs. Therefore, the dosimetric verifications performed in our study are valid at all stages of tumour growth.

## Supporting information

**S1 Data.** S1 File. Data file underlying the results presented in the manuscript. S2 File. CERR data file underlying the DVH calculations for the 0.5 cm diameter tumour. S3 File. CERR data file underlying the DVH calculations for the 0.75 cm diameter tumour. S4 File. CERR data file underlying the DVH calculations for the 1 cm diameter tumour.
(XLSX)

## Acknowledgments

The authors thank Dr. Magdalena Bazalova-Carter for valuable discussions on analysis of experiments and MC simulations.

## Author Contributions

**Conceptualization:** Amir Entezam, Davide Fontanarosa.

**Data curation:** Amir Entezam, Davide Fontanarosa.

**Formal analysis:** Amir Entezam, Andrew Fielding, David Bradley, Davide Fontanarosa.

**Investigation:** Amir Entezam.

**Methodology:** Amir Entezam, Andrew Fielding, David Bradley, Davide Fontanarosa.

**Project administration:** Andrew Fielding, Davide Fontanarosa.

**Software:** Amir Entezam, David Bradley.

**Supervision:** Andrew Fielding, Davide Fontanarosa.

**Validation:** Amir Entezam, Andrew Fielding, David Bradley, Davide Fontanarosa.

**Visualization:** Amir Entezam.

**Writing – original draft:** Amir Entezam, Davide Fontanarosa.

**Writing – review & editing:** Amir Entezam, Andrew Fielding, David Bradley.

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
