## [Decision Letter · Decision Letter 0]

10 May 2022

PONE-D-22-06377Absorbed Dose Calculation for a Realistic CT-derived Mouse Phantom Irradiated with a Standard Cs-137 Cell Irradiator Using a Monte Carlo MethodPLOS ONE

Dear Dr. Amir Entezam

Thank you for submitting your manuscript to PLOS ONE. After careful consideration, we feel that it has merit but does not fully meet PLOS ONE’s publication criteria as it currently stands. Therefore, we invite you to submit a revised version of the manuscript that addresses the points raised during the review process.

We look forward to receiving your revised manuscript.

Kind regards,

Mohamad Syazwan Mohd Sanusi

Academic Editor

PLOS ONE

Journal Requirements:

Reviewers' comments:

Reviewer's Responses to Questions

**Comments to the Author**

1. Is the manuscript technically sound, and do the data support the conclusions?

Reviewer #1: Yes

Reviewer #2: Yes

Reviewer #3: Yes

2. Has the statistical analysis been performed appropriately and rigorously? 

Reviewer #1: Yes

Reviewer #2: Yes

Reviewer #3: Yes

3. Have the authors made all data underlying the findings in their manuscript fully available?

Reviewer #1: No

Reviewer #2: Yes

Reviewer #3: Yes

4. Is the manuscript presented in an intelligible fashion and written in standard English?

Reviewer #1: Yes

Reviewer #2: Yes

Reviewer #3: Yes

5. Review Comments to the Author

Reviewer #1: Manuscript cogently describes a study based on the generation of a Monte Carlo model from a micro-CT image set

of a mouse with a xenograft tumour for absorbed dose investigations including tumour and out of field doses in the mouse.

1. Line 120-122 - justify the use of these size selections. Are these appropriate for mouse models?

2. Line 249 - 'countering' should perhaps be 'contouring'?

3. Line 261 - refers to 'our radio-immunotherapy investigations', please provide citations to this work.

4. Table 1 - was precision actually limited to only 1 decimal point for these mean doses? 100 mGy steps seem rather large. Perhaps report to 2 decimals (if measurement method supports). Also mean and range should perhaps be reported here?

Reviewer #2: I find this work to be accessible, well defined and nice to read. Below are some comments.

I think that figure 3 would benefit from adding a drawind which more clearly shows how the phantom and the film is located relative to the irradiation geometry.

I think table 1 should also include max values.

I would like an indication in the graph areas (or headers) of figure 8 that makes it immidiately obvious which graph respresents which tumour size.

Line 221: I would like you to include the equations for the gamma evaluation.

Line 227: I think 3mm and 3% is high, especially with respect to the dimensions of your experiment. I recommend that you redo your evaluation with 2mm and 2%, which I believe is also according to the latest recommendations from AAPM.

Line 279: Could you comment on the deviations between the dose curves of fig. 5?

Line 285: What are standard acceptance criteria for gamma test and how do your results relate to that?

Line 320: I believe you mean 8A, 8B and 8C.

Line 331-332: This refers to mean values. Update when you add max values.

Line 391: The challenge of homogeneous dose would be expected before the experiment. How come you did not include bolus as you suggest?

Line 413-414: This statement is true only within the range of studied tumour sizes. Either add this remark or weaken the statement of validity.

Reviewer #3: The paper entitled " Absorbed Dose Calculation for a Realistic CT-derived Mouse Phantom Irradiated with a Standard Cs-137 Cell Irradiator Using a Monte Carlo Method" has been reviewed for a potential publication in Journal PLOS ONE. The authors develop a CT-derived MC phantom generated from a mouse with a xenograft tumour that used to calculate both the dose heterogeneity in the tumour volume and out of field scattered dose for pre-clinical small animal irradiation experiments. This theoretical method is providing extensive data for the scientific community. Overall the paper is interesting. Most importantly, to ensure that the tumor is treated with the desired dose and that the dose to normal tissue (out of a field) has minimal impact. The titles along with their subtitles are prepared in a proper way.

I believe that the provided outcomes would enhance the existing data for preclinical radiobiological research.

However, I have the following recommendations for the authors, which will improve the overall quality of the paper.

* 16, 17 , 19, 21 , 22. 23, 24, 25, 26, 29, 30, 31, 38, 56, 57,58, 62 and 63 references need to be updated. They are all 2010 and older

6. PLOS authors have the option to publish the peer review history of their article (what does this mean?). If published, this will include your full peer review and any attached files.

Reviewer #1: No

Reviewer #2: No

Reviewer #3: No

---

## [Author Response · Author response to Decision Letter 0]

28 Sep 2022

We appreciate the editor and reviewers for taking the time to read and comment on our manuscript. The comments from the reviews are in black and our responses to the reviewers 1, 2, 3, 4 are highlighted in red, blue, green, and brown, respectively. Also, the changes that we have made to the manuscript are all in bold italic text in both response letter and revised manuscript. In addition to addressing the comments, our recently published study (in Medical Physics) was added to the reference list (see below) and has been cited in lines 102 and 135. 

46. Entezam A, Fielding A, Fontanarosa D. Characterization of a Novel Irradiation Method for Small Animal Radiotherapy Using a Monte Carlo Method. InMEDICAL PHYSICS 2022 Jun 1 (Vol. 49, No. 6, pp. E976-E976). 111 RIVER ST, HOBOKEN 07030-5774, NJ USA: WILEY.

Reviewer #1: Manuscript cogently describes a study based on the generation of a Monte Carlo model from a micro-CT image set of a mouse with a xenograft tumour for absorbed dose investigations including tumour and out of field doses in the mouse.

1. Line 120-122 - justify the use of these size selections. Are these appropriate for mouse models?

Thank you for the valuable comments. Yes, the selected voxel size is the most appropriate for the mouse model in our study. The following statement that explains the reason for the size selection was added to the manuscript (line 123-126). Also, reference number 51 was added to the manuscript and reference numbers were updated throughout the manuscript accordingly.

“There is a trade-off between voxel size and statistical accuracy for MC calculations in small animal [32]. The selected voxel size in this study allows for sufficient statistical accuracy while providing adequate resolution for small animal phantoms [32], [51].”

2. Line 249 - 'countering' should perhaps be 'contouring'? 

The word was corrected in line 262 as suggested. 

3. Line 261 - refers to 'our radio-immunotherapy investigations', please provide citations to this work.

The first paper for our radio-immunotherapy study is currently still under preparation. However, our previous physics study provides adequate information on the requested citation. So, we provided citations to our previous physics work (reference number 44) in line 274.

4. Table 1 - was precision actually limited to only 1 decimal point for these mean doses? 100 mGy steps seem rather large. Perhaps report to 2 decimals (if measurement method supports). Also mean and range should perhaps be reported here?

Thank you for the comment. We agree with the reviewer that mean doses should be reported to 2 decimals. So, in the revised version, the mean doses in Table 2 are reported with 2 decimals (see below). Also, to include the range, the minimum and maximum dose results are reported in table 2 to present the lower and upper range limits of the doses. 

OARs Minimum dose (Gy)

Tumour diameter (cm)

 0.5 0.75 1 Maximum dose (Gy)

Tumour diameter (cm)

0.5 0.75 1 Mean dose (Gy)

 Tumour diameter (cm)

 0.5 0.75 1

Rlung 0.73 0.72 0.75 1.35 1.38 1.36 1.10 1.20 1.21

Llung 0.84 0.85 0.79 1.36 1.37 1.33 1.20 1.21 1.13

Spine 0.92 0.95 0.94 1.37 1.36 1.29 1.14 1.24 1.18

Heart 0.90 0.95 0.88 1.35 1.32 1.34 1.22 1.23 1.20

Liver 0.94 0.87 0.95 1.34 1.33 1.38 1.21 1.20 1.22

Lkidney 0.92 0.85 0.95 1.33 1.29 1.32 1.19 1.16 1.21

Rkidney 0.89 0.85 0.95 1.36 1.31 1.37 1.20 1.18 1.22

Tumour 12.22 12.05 12.57 20.67 20.53 20.73 16.75 16.92 17.22

Reviewer #2: I find this work to be accessible, well defined, and nice to read. Below are some comments.

1. I think that figure 3 would benefit from adding a drawind which more clearly shows how the phantom and the film is located relative to the irradiation geometry.

Thank you for the valuable comments. A cutaway drawing of the phantom irradiation was added to figure 3 as below and the description was given in the caption (D) in lines 211-214.

“D) Cutaway drawing of the collimator with the PMMA phantom positioned at the midplane of the collimator and the film dosimeter sandwiched between phantom slabs irradiated with Cs-137 beam. Film dosimeter, phantom slabs, and the collimator’s components are marked.”

2. I think table 1 should also include max values.

We agree that the maximum dose values should be reported. This was requested by the reviewer 1 too. So, the maximum dose values have been already added to table 2. 

3. I would like an indication in the graph areas (or headers) of figure 8 that makes it immediately obvious which graph represents which tumour size.

Headers with tumour sizes were added to the graph and figure 8 was updated in the manuscript (below). 

Line 221: I would like you to include the equations for the gamma evaluation.

The gamma index equation along with its explanation (below) were added to the manuscript in line 232-237. 

“…quantity through the following equation, known as the Gamma value [56]. 

γ(r⃗_ref,r⃗_m )=〖min 〗⁡{√((|〖r⃗_ref-r⃗_m |〗^2)/〖DTA〗^2 + 〖〖|D (r⃗〗_ref)-D(r⃗_m)|〗^2/(ΔD^2 ))} (1)

where

〖|D (r⃗〗_ref)-D(r⃗_m)| is the dose difference and 〖|r⃗〗_ref-r⃗_m | is the distance between the dose points. “

Line 227: I think 3mm and 3% is high, especially with respect to the dimensions of your experiment. I recommend that you redo your evaluation with 2mm and 2%, which I believe is also according to the latest recommendations from AAPM.

We agree with the reviewer. The gamma analysis has been performed also for 2 mm and 2% limits and the corresponding gamma analysis plot (Fig 5) has been updated in the manuscript (shown below). 

Line 279: Could you comment on the deviations between the dose curves of fig. 5?

To address the comment the following statement was added to the manuscript in lines 298-300. (Please note that the corresponding detailed discussion is also present in the discussion section).

“The major deviations between the dose profiles might arise from the statistical nature of the MC simulation as well as uncertainties in the GAFchromic film dosimetry as detailed in the discussion section.”

Line 285: What are standard acceptance criteria for gamma test and how do your results relate to that?

Gamma values of less than 1 imply that the gamma test was passed. This has been already mentioned in the material and method section. This was restated in the result section lines 297-299 to address the comment as below:

“Only two gamma values were smaller than 1 and therefore failed the gamma test.” 

Line 320: I believe you mean 8A, 8B and 8C.

This was corrected in line 335 following the comment. 

Line 331-332: This refers to mean values. Update when you add max values.

The values were updated in lines: 42, 347, 396, 426, 431, 461. 

Line 391: The challenge of homogeneous dose would be expected before the experiment. How come you did not include bolus as you suggest?

To explain why the bolus was not included for irradiation in this study the following statement was added to the discussion section in lines 420 to 424. Also, two new references (64 and 65) were included to support the statement. 

“It must be highlighted that our specific study ultimately aims at investigating abscopal effects of RT [45], for which we do not need to deliver a homogeneous dose to the whole target [44], but only deliver enough dose to a certain fraction of the target [64], [65]. So, at this stage, it was not essential to employ a bolus for tumour irradiations in our experiments.”

Line 413-414: This statement is true only within the range of studied tumour sizes. Either add this remark or weaken the statement of validity.

The remark recommended by the reviewer was added to the manuscript in lines 435-437. 

“Based on this, the dosimetric verifications performed in the experiments are valid at all stages of mice irradiations, when tumours grow within the range of the investigated tumour dimensions.” 

Reviewer #3: The paper entitled " Absorbed Dose Calculation for a Realistic CT-derived Mouse Phantom Irradiated with a Standard Cs-137 Cell Irradiator Using a Monte Carlo Method" has been reviewed for a potential publication in Journal PLOS ONE. The authors develop a CT-derived MC phantom generated from a mouse with a xenograft tumour that used to calculate both the dose heterogeneity in the tumour volume and out of field scattered dose for pre-clinical small animal irradiation experiments. This theoretical method is providing extensive data for the scientific community. Overall the paper is interesting. Most importantly, to ensure that the tumor is treated with the desired dose and that the dose to normal tissue (out of a field) has minimal impact. The titles along with their subtitles are prepared in a proper way. 

I believe that the provided outcomes would enhance the existing data for preclinical radiobiological research. 

However, I have the following recommendations for the authors, which will improve the overall quality of the paper.

* 16, 17 , 19, 21 , 22. 23, 24, 25, 26, 29, 30, 31, 38, 56, 57,58, 62 and 63 references need to be updated. They are all 2010 and older. 

Thank you for the valuable comment which helps us to improve the overall quality of the paper. The intended references were updated to 2010 and more recent publications in the manuscript as follows:

16. Xie T, Zaidi H. Development of computational small animal models and their applications in preclinical imaging and therapy research. Medical Physics. 2016 Jan;43(1):111-31.

17. Kainz W, Neufeld E, Bolch WE, Graff CG, Kim CH, Kuster N, Lloyd B, Morrison T, Segars P, Yeom YS, Zankl M. Advances in computational human phantoms and their applications in biomedical engineering—a topical review. IEEE transactions on radiation and plasma medical sciences. 2018 Dec 3;3(1):1-23.

19. Xu XG. An exponential growth of computational phantom research in radiation protection, imaging, and radiotherapy: a review of the fifty-year history. Physics in Medicine & Biology. 2014 Aug 21;59(18):R233.

21. Xie T, Zaidi H. Development of computational small animal models and their applications in preclinical imaging and therapy research. Medical Physics. 2016 Jan;43(1):111-31.

22. Xie T, Zaidi H, Segars P. Design and construction of computational animal models. InComputational Anatomical Animal Models: Methodological developments and research applications 2018 Dec 1. IOP Publishing.

23. Lee CL, Park SJ, Jeon PH, Jo BD, Kim HJ. Dosimetry in small-animal CT using Monte Carlo simulations. Journal of Instrumentation. 2016 Jan 27;11(01):T01003.

24. Papadimitroulas P, Kagadis GC. Applications of computational animal models in ionizing radiation dosimetry. InComputational Anatomical Animal Models: Methodological developments and research applications 2018 Dec 1. IOP Publishing.

25. Perrot Y, Maigne L, Breton V, Donnarieix D. Validation of GATE 6.1 for targeted radiotherapy of metastic melanoma using 131I-labeled benzamide. InThird European Workshop on Monte Carlo Treatment Planning 2012 May 15 (pp. 5-p).

26. Vasquez SX, Shah N, Hoberman AM. Small animal imaging and examination by micro-CT. InTeratogenicity testing 2013 (pp. 223-231). Humana Press, Totowa, NJ.

29. Praveenkumar RD, Santhosh KP, Augustine A. Estimation of inhomogenity correction factors for a Co-60 beam using Monte Carlo simulation. Journal of Cancer Research and Therapeutics. 2011 Jul 1;7(3):308.

30. De Marzi L, Lesven C, Ferrand R, Sage J, Boulé T, Mazal A. Calibration of CT Hounsfield units for proton therapy treatment planning: use of kilovoltage and megavoltage images and comparison of parameterized methods. Physics in Medicine & Biology. 2013 May 29;58(12):4255.

31. Schreuder AN, Bridges DS, Rigsby L, Blakey M, Janson M, Hedrick SG, Wilkinson JB. Validation of the RayStation Monte Carlo dose calculation algorithm using realistic animal tissue phantoms. Journal of Applied Clinical Medical Physics. 2019 Oct;20(10):160-71.

38. Wiklund K, Toma-Dasu I, Lind BK. The influence of dose heterogeneity on tumour control probability in fractionated radiation therapy. Physics in Medicine & Biology. 2011 Nov 15;56(23):7585.

59. Niko H, Dafina X, Theodhor K, Ervis T. Calculation Methods in Radiotherapy Using MATLAB. Journal International Environmental Application Science, ISSN. 2014 Jun 1:1307-0428.

60. Baiker M, Milles J, Dijkstra J, Henning TD, Weber AW, Que I, Kaijzel EL, Löwik CW, Reiber JH, Lelieveldt BP. Atlas-based whole-body segmentation of mice from low-contrast Micro-CT data. Medical image analysis. 2010 Dec 1;14(6):723-37.

67. Formenti SC. Optimizing dose per fraction: a new chapter in the story of the abscopal effect?. International Journal of Radiation Oncology, Biology, Physics. 2017 Nov 1;99(3):677-9.

68. Drissi R, Wu J, Hu Y, Bockhold C, Dome JS. Telomere shortening alters the kinetics of the DNA damage response after ionizing radiation in human cells. Cancer Prevention Research. 2011 Dec;4(12):1973-81.

Reviewer #4: The paper is well written and structured and the conclusions are well supported by the experimental results and calculations. However, I think that the authors should be more stringent about the radiation dose quantity used. The term “dose” is used in the manuscript and this ought to be replaced by “absorbed dose”.

Thank you for the comments. To address this comment, the term “dose” was replaced by “absorbed dose” in lines: 39, 83, 130, 316, 346, 363, 365, 402, 409 throughout the manuscript.

Specific comments:

Line 247: The second “right lung” should be “right kidney”.

“right lung” was replaced by “right kidney” in Line 260. 

Line 320: I guess that it should be Figure 8A, 8B, 8C.

This was corrected in line 335 following the comment.

---

## [Decision Letter · Decision Letter 1]

13 Oct 2022

PONE-D-22-06377R1Absorbed Dose Calculation for a Realistic CT-derived Mouse Phantom Irradiated with a Standard Cs-137 Cell Irradiator Using a Monte Carlo MethodPLOS ONE

Dear Dr. Entezam,

Thank you for submitting your manuscript to PLOS ONE. After careful consideration, we feel that it has merit but does not fully meet PLOS ONE’s publication criteria as it currently stands. Therefore, we invite you to submit a revised version of the manuscript that addresses the points raised during the review process.

We look forward to receiving your revised manuscript.

Kind regards,

Mohamad Syazwan Mohd Sanusi

Academic Editor

PLOS ONE

Journal Requirements:

Additional Editor Comments:

Dear author, I would like to invite you to complete the revision based on the missing comments by Reviewer #4:

General comments:

In this article the authors present a method to evaluate both – the dose heterogeneity in the tumour

volume and the out of field scattered dose for pre-clinical small animal irradiation experiments

using computed tomography derived Monte Carlo phantoms – in this particular instance – a

phantom of a mouse with a xenographic tumour. The authors built a virtual mouse phantom from a

whole-body micro-CT image set of a mouse with 1 cm diameter xenographic tumour, which was

then irradiated using a BEAMnrc Monte Carlo user code. Then, doses within the mouse phantom

and dose volume histograms were calculated and displayed, allowing a dosimetric verification of

actual tumour bearing mouse irradiation to be performed.

The study is generally well written and concise, however, there are few mistakes and some other

issues that needs to be addressed. Please see my comments as attached and make changes to the

current version of the manuscript.

I recommend minor revision of the manuscript before being considered for publication in PLOS

ONE.

Comments:

1. Lines 180-206; Due to the nature of complexity of the mouse phantom, which uses different

materials representing the air, lung, soft tissue and bone materials within the phantom, the choice of

a single, simple PMMA phantom for verification could be interpreted as a bit weak in my opinion.

Especially, as the PMMA material is not utilised in the actual mouse phantom. I would suggest

either finding an additional physical phantom of at least of one of the materials used within the

mouse phantom for additional verification, or using other means to strengthen the sufficiency of

verification using only PMMA phantom – for example references.

2. Lines 225; Why not using a simpler way to analyse the data? Either way, the data obtained from

the Gafchromic film is 2D, so just comparing 2D dose maps with some particular selected values

would be enough, as in this 2D situation, the choice of the Gamma index analysis does not seem to

provide significant advantages over a simple 2D dose map comparison. Although, if there are such

advantages, this should be mentioned in the running text.

3. In a case, where the Gamma index analysis is sufficiently substantiated by the authors, I believe

there should be somewhat more explanation within the running text about the Gamma index. At

least some adapted figures explaining the basic principles of the method, which then the reference

to the work by Diamantopoulos et al. would complete.

4. Line 295; All doses were normalised to the depth of maximum dose? Following text suggestthat

the Dmax means the maximum dose. Please clarify.

5. Lines 317-329; Differences within the different DVH curves are discussed in the running text. It

would be much clearer if the curves would be plotted on the same graph, so that the differences

would be more apparent to the reader.

6. Fig 2, mistakes within the text in figure, like “Virtual Cs-137 sorce” and “used at beam angel”.

Please correct.

Reviewers' comments:

Reviewer's Responses to Questions

**Comments to the Author**

1. If the authors have adequately addressed your comments raised in a previous round of review and you feel that this manuscript is now acceptable for publication, you may indicate that here to bypass the “Comments to the Author” section, enter your conflict of interest statement in the “Confidential to Editor” section, and submit your "Accept" recommendation.

Reviewer #1: All comments have been addressed

Reviewer #2: All comments have been addressed

Reviewer #5: All comments have been addressed

2. Is the manuscript technically sound, and do the data support the conclusions?

Reviewer #1: (No Response)

Reviewer #2: Yes

Reviewer #5: Yes

3. Has the statistical analysis been performed appropriately and rigorously? 

Reviewer #1: (No Response)

Reviewer #2: Yes

Reviewer #5: Yes

4. Have the authors made all data underlying the findings in their manuscript fully available?

Reviewer #1: (No Response)

Reviewer #2: Yes

Reviewer #5: Yes

5. Is the manuscript presented in an intelligible fashion and written in standard English?

Reviewer #1: (No Response)

Reviewer #2: Yes

Reviewer #5: Yes

6. Review Comments to the Author

Reviewer #1: (No Response)

Reviewer #2: I find all my comments appropriately addressed and suggest acceptance of the article for publication.

Reviewer #5: (No Response)

7. PLOS authors have the option to publish the peer review history of their article (what does this mean?). If published, this will include your full peer review and any attached files.

Reviewer #1: No

Reviewer #2: **Yes: **Dr Jakob H. Lagerlöf

Reviewer #5: No

---

## [Author Response · Author response to Decision Letter 1]

24 Oct 2022

We appreciate the editor and the reviewer for taking the time to read and comment on our manuscript. The comments from the reviewer-4 are in black and our responses are highlighted in red. Also, the changes that we have made to the manuscript are all in bold italic text in both response letter and the revised manuscript. With the exceptions of exchanging the reference numbers 56 and 57 and correcting the reference number 6 in the text, we did not make any changes to the reference list. 

Please note that this is the second letter of “Response to Reviewer” for our submission. In the first round of the revision (with decision of “MINOR REVISION”), we addressed all comments from the reviewers 1, 2, 3, and 5. So, in the present revision only the comments of reviewer 4 have been addressed.

Comments

 Lines 180-206; Due to the nature of complexity of the mouse phantom, which uses different materials representing the air, lung, soft tissue and bone materials within the phantom, the choice of a single, simple PMMA phantom for verification could be interpreted as a bit weak in my opinion. Especially, as the PMMA material is not utilised in the actual mouse phantom. I would suggest either finding an additional physical phantom of at least of one of the materials used within the mouse phantom for additional verification, or using other means to strengthen the sufficiency of verification using only PMMA phantom – for example references.

Thank you for the valuable comments which helps us to improve the paper. 

Verification of the Monte Carlo modelling is commonly fulfilled by comparing the measured and calculated dose distributions. Many studies use dosimeters within “slabs of PMMA phantoms” or “simple water phantoms” to measure the doses and compare them with the MC calculated results. This indicates that the utilization of “simple PMMA phantoms” for dose measurements and comparing the measured doses with the MC calculated results is a valid verification method as we used in this study. To address the comment, the following statement was added to the discussion section (lines 379-382) and the relevant references were provided (as suggested by the reviewer). 

“It is worth mentioning that several studies have used the same method for verifying MC simulations [55], [32], [51], which confirms the validity of the verification method used in this study, i.e., comparing the doses measured within a PMMA phantom with the corresponding calculated MC doses.”

 Why not using a simpler way to analyse the data? Either way, the data obtained from

the Gafchromic film is 2D, so just comparing 2D dose maps with some particular selected values would be enough, as in this 2D situation, the choice of the Gamma index analysis does not seem to provide significant advantages over a simple 2D dose map comparison. Although, if there are such advantages, this should be mentioned in the running text. 

Since Gamma analysis is the gold standard for dose distribution comparisons, authors decided to perform the gamma index analysis for a robust MC simulation verification in this study. So, to address the comment, the advantage of the gamma index analysis along with the relevant reference to support the statement were added to the manuscript in lines 228-231 as follows:

“Gamma analysis is the most effective method for comparing two dose distributions [56]. It is established as the gold standard in verification procedures among all available methods and clinical decisions are made based on its outcomes [56], [57]. “ 

 In a case, where the Gamma index analysis is sufficiently substantiated by the authors, I believe there should be somewhat more explanation within the running text about the Gamma index. At least some adapted figures explaining the basic principles of the method, which then the reference to the work by Diamantopoulos et al. would complete.

The complete equation of the gamma and its corresponding explanations (see equation 1 below) were added to the manuscript based on comment from the reviewer 2 in the first round of revision submission as follows: 

unitless quantity through the following equation, known as the Gamma value [56]. 

γ(r⃗_ref,r⃗_m )=〖min 〗⁡{√((|〖r⃗_ref-r⃗_m |〗^2)/DTA^2 + 〖〖|D (r⃗〗_ref)-D(r⃗_m)|〗^2/(ΔD^2 ))} (1)

where

〖|D (r⃗〗_ref)-D(r⃗_m)| is the dose difference and 〖|r⃗〗_ref-r⃗_m | is the distance between the dose points. 

To address the comment, the following explanations were added to the manuscript (lines 240-243, next to the Gamma equation) to provide more details on the basic principles of the Gamma analysis as requested.

“In this way, an acceptance region (an ellipse) is created around every single point of the reference dose distribution. This ellipse should compass the evaluated dose point to pass the gamma test. Mathematically, gamma values of less than 1 imply that the gamma test was passed [56].”

4. Line 295; All doses were normalised to the depth of maximum dose? Following text suggest that the Dmax means the maximum dose. Please clarify.

Thank you for the significant comment. Dmax is maximum dose. The text was corrected following the comment in line 315.

5. Lines 317-329; Differences within the different DVH curves are discussed in the running text. It would be much clearer if the curves would be plotted on the same graph, so that the differences would be more apparent to the reader.

We agree with the reviewer that plotting all DVHs on the same graph contributes to the clarity. So, to address the comment, the DVHs were plotted on a single graph and figure 8 was updated in the manuscript (see below). Also, the figure number were updated in the manuscript (in lines 340, 356, 357) and the caption of the figure 8 was modified in lines 359-361. 

“Fig 8. DVHs plots. DVHs of the tumours and OARs for 0.5 cm, 0.75 cm, and 1 cm tumour irradiations are demonstrated in dash, dot, and solid lines, respectively. DVHs of the tumours are in black and colours of the DVHs for OARs are presented in the graph.”

6. Fig 2, mistakes within the text in figure, like “Virtual Cs-137 sorce” and “used at beam angel”.

Thank you for your scrutiny. Mistakes in the figure were corrected and the figure 2 was updated in the manuscript as below.

---

## [Editor Report · Decision Letter 2]

11 Nov 2022

PONE-D-22-06377R2Absorbed Dose Calculation for a Realistic CT-derived Mouse Phantom Irradiated with a Standard Cs-137 Cell Irradiator Using a Monte Carlo MethodPLOS ONE

Dear Dr. Entezam,

Thank you for submitting your manuscript to PLOS ONE. After careful consideration, we feel that it has merit but does not fully meet PLOS ONE’s publication criteria as it currently stands. Therefore, we invite you to submit a revised version of the manuscript that addresses the points raised during the review process. Please ensure that your decision is justified on PLOS ONE’s publication criteria and not, for example, on novelty or perceived impact.

Please submit your revised manuscript by Dec 26 2022 11:59PM If you will need more time than this to complete your revisions, please reply to this message or contact the journal office at plosone@plos.org. Please include the following items when submitting your revised manuscript:A rebuttal letter that responds to each point raised by the academic editor and reviewer(s). You should upload this letter as a separate file labeled 'Response to Reviewers'.A marked-up copy of your manuscript that highlights changes made to the original version. You should upload this as a separate file labeled 'Revised Manuscript with Track Changes'.An unmarked version of your revised paper without tracked changes. You should upload this as a separate file labeled 'Manuscript'.If applicable, we recommend that you deposit your laboratory protocols in protocols.io to enhance the reproducibility of your results. Protocols.io assigns your protocol its own identifier (DOI) so that it can be cited independently in the future. For instructions see: https://journals.plos.org/plosone/s/submission-guidelines#loc-laboratory-protocols. Additionally, PLOS ONE offers an option for publishing peer-reviewed Lab Protocol articles, which describe protocols hosted on protocols.io. Read more information on sharing protocols at https://plos.org/protocols?utm_medium=editorial-email&utm_source=authorletters&utm_campaign=protocols.

We look forward to receiving your revised manuscript.

Kind regards,

Mohamad Syazwan Mohd Sanusi

Academic Editor

PLOS ONE

Journal Requirements:

Additional Editor Comments:

Thanks to the authors for the revised mansucript. Before we proceed to accept the submited work for publications, there are few minor corrections are needed to improve the quality of the submitted work. Kinldy find my following comments for your references.

Title: ok

Abstract:sufficient, Line 33 please name the phantom.

Line 94-97:This is the specific objectives of the work and in Line 101-103 again, the author mentioned the general motivation of this work. Please revise those lines to make avoid redundant and confused the reader.

Line 98-101: This should be in methodology to address Pb colimator consideration for efficient irradiations. Please remove from Introduction.

Line 104-105: This is the important aspect that need to be focus in introduction. Throughout the Introduction Line 48-85, the authors are addressing the history of mc and ct-based ocmputational phantom, importance of absosrbed dose investigations,

problems statetement ie. no available dosimetry study on xenographic tumour dimensions, literature works. Based on the given title, the author should highlight the literature review on how the experimental verification of dosimetry on

the proposed phantom was conducted in the past?

Line 105-109: The importance or the finding from this study should be in early part of introduction not in the last para of introduction.

Line 168-170: Please elaborate the importance of these cut-offs configutrations to MC computations.

Under 2.2.2 & 2.3.1: Please include the desxriptoion of secondary electrons transports and other photon-electron interaction treatments for MC computation.

Line 180: Please extend the discussion why PMMA is chosen over other tissue weighted equivalent materials i.e nylon, polyacetate? Indicate the composition, density and compare to other tissue eg bone,

adipose, muscle etc. as well as reference study that using PMMA for dosimetry verifcation of MC phantom study. Any approximation need to be stated in this part fopr the considertation of PMMA.

Under 2.3.1, Line 190 : Please extend the discussion on the approach and technique for quality control of Gafchromic EBT3 film measurement? any other solid state dosimeters used for dose corrections?

Result, 3.1: Please include a table of data for Figure 5A. The given tabulated data will benefit the reader apart of it is easier for evaluator to check the disparity of dose over x for both MC data vs Exp. PMMA doses. Im wondering the variation of the error of MC computations for different parameters, not sure I have mislook on the error figure in discussion. Please include the value in this table as well as the formula for relative errors estimations.

Discussion: sufficient

Conclusion: sufficient

---

## [Author Response · Author response to Decision Letter 2]

19 Dec 2022

We appreciate the Editor for taking the time to read and comment on our manuscript. The comments from the Editor are in black and our responses are highlighted in red. Also, the changes that we have made to the manuscript are all in bold italic text in both response letter and the revised manuscript. 

Reference 32, 50, 56, 65, 66, 76, 68, 69, 70, 71, 72, and 73 were added to the manuscript. Accordingly, reference numbers 39-82 were updated throughout the manuscript. 

Please note that this is the third letter of “Response to Reviewers” for our submission. In the last rounds of the revisions (with decision of “MINOR REVISION”), we addressed all comments from the reviewers 1, 2, 3, 4, and 5. So, in the present revision only the comments of the Editor have been addressed.

Additional Editor Comments:

Thanks to the authors for the revised mansucript. Before we proceed to accept the submited work for publications, there are few minor corrections are needed to improve the quality of the submitted work. Kinldy find my following comments for your references.

1. Line 33 please name the phantom.

Thank you for the valuable comments which help us improve the paper. The name of the phantom has been added in line 33. 

2. Line 94-97: This is the specific objectives of the work and in Line 101-103 again, the author mentioned the general motivation of this work. Please revise those lines to make avoid redundant and confused the reader.

The redundant information in lines 101-103 has been removed.

3. Line 98-101: This should be in methodology to address Pb colimator consideration for efficient irradiations. Please remove from Introduction.

The details of the collimator have been removed from the Introduction as suggested and have been added to the methodology section with minor changes as below (lines 136-142).

“We have previously constructed an add-on lead collimator to allow for targeted irradiation of mice [51]. In previous studies, we showed that our collimator combined with a 137Cs source irradiator is an effective method for irradiating small animal xenograft tumours [51], [52], [53]. The simulation of MC mouse phantom irradiation was performed using a 1-cm diameter beam produced by the collimator [51] mounted onto a Gammacell 40 Exactor irradiator unit (Best Theratronics, ON K2K OE4, Canada) located at Transitional Research Institute (TRI), Brisbane, Australia.”

4. Line 105-109: The importance or the finding from this study should be in early part of introduction not in the last para of introduction. 

Thank you for the important comment. The importance of the study has been moved to an earlier part of the Introduction section (lines 89-94). 

5. Line 168-170: Please elaborate the importance of these cut-offs configutrations to MC computations.

Thank you for the comment. Based on the literature, the selected cut-offs allow for an efficient dose calculation while reducing the CPU calculation time. To address the comment the following explanation along with the relevant references were added to the manuscript in lines 180-183. 

“These well-established cut-off settings allow for efficient dose calculations while reducing the CPU calculation time [50]. “

6. Under 2.2.2 & 2.3.1: Please include the description of secondary electrons transports and other photon-electron interaction treatments for MC computation.

Thank you for the important comment. The missing information about the secondary electrons transport and photon-electron interactions are provided in lines 173-181 and 236-237 under 2.2.2 and 2.3.2, respectively, as suggested by the Editor.

lines 173-180

“Electron impact ionization, atomic relaxations, and low energy photon interactions, such as Rayleigh scattering and bound Compton scattering, have been included in our MC simulations [55], [56]. Also, through the condensed history technique, catastrophic inelastic scattering and bremsstrahlung were taken into account in the transport of secondary electrons [56]. Global Electron transport cut-offs (ECUT) and Global photon transport cut-offs (PCUT) were 0.521 MeV and 0.01 MeV respectively [54]. Also, the low energy thresholds for electrons and photons AE and AP were set to 0.521 and 0.01, consistent with the ECUT and PCUT parameters, respectively.” 

lines 236-237

“The interactions explained in section 2.2.2 were also included in DOSXYZnrc simulations [58]." 

7. Line 104-105: This is the important aspect that need to be focus in introduction. Throughout the Introduction Line 48-85, the authors are addressing the history of mc and ct-based ocmputational phantom, importance of absosrbed dose investigations,problems statetement ie. no available dosimetry study on xenographic tumour dimensions, literature works. Based on the given title, the author should highlight the literature review on how the experimental verification of dosimetry onthe proposed phantom was conducted in the past?

As suggested by the Editor, to highlight the literature review on how the verification of MC modelling/simulations has been conducted in the past we performed a systematic literature review, using the PRISMA diagram (shown below). 

Systematic PRISMA literature review details

1. Database: PubMed 

Keyword: “PMMA phantom” AND "dose measurement" AND Monte Carlo AND (radiotherapy OR Irradiation) AND “small animal”. This includes all the search terms in the different combinations using Boolean operators (AND or OR). 

Main limits: (years of search >2000, English language).

2. Articles that appear more than once were manually removed.

3. Titles and abstracts for articles that are relevant to this research were screened. All 28 articles appeared to provide information related to the research. As a result, 0 articles excluded based on this screening. 

4. After reading them, 13 articles not exactly relevant to the context were excluded. 

5. As the final step, we subtracted the number of excluded articles or records during the eligibility review of full texts from the total number of articles reviewed for eligibility. Finally, the PRISMA flow diagram completed with "15" articles for literature review. Only 6 studies presented exactly the same method of PMMA irradiations for the MC simulation verifications. Explanations about these 6 studies are included in the manuscript. 

15 articles on the small animal phantom dosimetry using MC simulations have been systematically reviewed. The literature review gives rise to the fact that verification of the Monte Carlo modelling is commonly fulfilled by comparing the measured and calculated dose distributions using “slabs of PMMA phantoms” or “simple water phantoms. Similar to our study, 6 studies have used dosimeters within “slabs of PMMA phantoms” to measure the doses and compare them with the MC calculated results. To address the comment, the brief literature review was added to the introduction (lines 104-110) and the relevant references were provided (as suggested by the Editor) as below. 

“Verification of Monte Carlo simulations is commonly accomplished by comparing dose distributions obtained from physical dose measurements with the corresponding calculated doses. Good agreement between the measured and calculated doses guarantees the accuracy of MC modelling. Several studies have used dosimeters within slabs of PMMA phantom or simple water phantoms to measure the doses, comparing them with the MC simulation results [47], [48], [49], [50], [51], [52]. In this study, we use the same method for the verification of our MC simulations.”

8. Line 180: Please extend th e discussion why PMMA is chosen over other tissue weighted equivalent materials i.e nylon, polyacetate? Indicate the composition, density and compare to other tissue eg bone, adipose, muscle etc. as well as reference study that using PMMA for dosimetry verifcation of MC phantom study. Any approximation need to be stated in this part fopr the considertation of PMMA.

Thank you for the comment. Firstly, as it was shown in our systematic literature review, several studies used PMMA phantom irradiations to verify MC modelling. This indicates that the utilization of “simple PMMA phantoms” for dose measurements and comparing the measured doses with the MC calculated results is a valid verification method as we used in this study.

Secondly, PMMA is chosen over other materials because it is tissue equivalent, has a reproducible composition, and easy to machine. Also, PMMA phantoms are employed for standard dosimetry practices like calibration of in-phantom dose rate (relevant references were given). The PMMA phantom well mimics the soft tissue composition and therefore it effectively approximates the tissue of the xenograft tumour. 

Thirdly, one reference was given in order to provide requested information on the PMMA phantom composition and electron density. 

Lastly, one reference was given in which the mouse tissues were considered PMMA equivalent for dosimetric verification purposes.

To address the comment the following statement is added to the discussion section in lines 401-413 as below: 

“As mentioned in the introduction section, several studies have used the same method for verifying MC simulations. This provides strong support for the validity of the verification method used in this study, i.e., comparing the doses measured within a PMMA phantom with the corresponding calculated MC doses. It is worth mentioning that we used PMMA material in this study because it is tissue equivalent, has a reproducible composition, is easy to machine, and it is commonly used for radiation dosimetry investigations [65], [66], [67], [68], [69]. Also, PMMA phantoms are typically employed for standard dosimetry practices such as calibration of in-phantom dose rate [70], [71]. PMMA is found to be an excellent soft-tissue substitute as its material properties such as effective atomic number and electronic density are close to that of soft tissue [72]. So, the in-house built PMMA phantom well mimics soft tissue composition of the mouse. Similar to our method, in research conducted by Kuess et al [73] the mouse tissues were considered PMMA equivalent for dosimetric verification purposes.” 

9. Under 2.3.1, Line 190 : Please extend the discussion on the approach and technique for quality control of Gafchromic EBT3 film measurement? any other solid state dosimeters used for dose corrections?

Thank you for the comment. To ensure the accuracy of the Gafchromic EBT3 film measurement, film dosimetry was performed based on the Gafchromic film dosimetry procedure given in previous studies (as has been already mentioned in section 2.3.1 lines 213-214. Subsequently, the measured doses were compared against the corresponding doses obtained from MOSKIN dosimeter for dose corrections. To address the comment the following statement was added to the discussion section line 425-428. 

“To ensure the accuracy of our Gafchromic EBT3 film measurement, the measured doses were compared against the corresponding doses obtained from MOSKIN dosimeter as detailed in our previous study [51].”

10. Result, 3.1: Please include a table of data for Figure 5A. The given tabulated data will benefit the reader apart of it is easier for evaluator to check the disparity of dose over x for both MC data vs Exp. PMMA doses. Im wondering the variation of the error of MC computations for different parameters, not sure I have mislook on the error figure in discussion. Please include the value in this table as well as the formula for relative errors estimations.

A table containing the measured and calculated data was included to the manuscript to demonstrate the disparity of doses for each point of MC calculation and the corresponding measured points (see below). Also, the errors for all calculated points were obtained from the 3ddose file and added to the table. The following statement along with the table was added to the manuscript (lines 311-314). 

“The MC calculated doses for all points along with the corresponding measured doses, corresponding to the profiles shown in figure 5A, are presented in Table 1. Also, the errors for all MC calculated doses were obtained from the 3ddose file and presented in the last column of the table.”

Table 1. Measured and calculated doses and MC dose calculation errors. 

Distance (cm) Measured doses (%) MC Calculated doses (%) MC calculation errors (%)

0 100 100 0.88

0.12 99.44 97.53 0.79

0.22 97.62 98.13 0.65

0.32 94.05 99.75 1.18

0.42 82.37 90.42 0.78

0.52 62.30 62.64 0.85

0.62 34.74 35.31 1.62

0.72 15.73 14.32 1.92

0.82 13.02 11.85 2.10

0.92 10.57 10.61 2.92

1.02 10.45 11.12 2.79

1.12 11.20 11.48 3.29

1.22 9.61 9.82 2.91

1.32 9.60 9.01 3.22

1.42 9.66 8.61 2.99

1.52 9.09 8.73 2.82

1.62 9.60 8.67 3.30

1.72 9.50 8.94 1.98

1.82 9.38 9.02 3.21

1.92 9.43 8.92 2.96

2.02 9.46 7.39 3.34

2.12 9.54 7.40 2.92

2.22 9.41 7.13 3.07

2.32 9.32 6.82 2.82

2.50 9.39 6.60 2.98

Dose values for all points of the calculated profile along with the corresponding measured doses (normalized to their respective maximum doses). Dose calculation errors are presented in the last column.

---

## [Editor Report · Decision Letter 3]

8 Jan 2023

Absorbed Dose Calculation for a Realistic CT-derived Mouse Phantom Irradiated with a Standard Cs-137 Cell Irradiator Using a Monte Carlo Method

PONE-D-22-06377R3

Dear Dr. Entezam,

We’re pleased to inform you that your manuscript has been judged scientifically suitable for publication and will be formally accepted for publication once it meets all outstanding technical requirements.

Kind regards,

Mohamad Syazwan Mohd Sanusi

Academic Editor

PLOS ONE

---

## [Editor Report · Acceptance letter]

16 Jan 2023

PONE-D-22-06377R3 

Absorbed Dose Calculation for a Realistic CT-derived Mouse Phantom Irradiated with a Standard Cs-137 Cell Irradiator Using a Monte Carlo Method 

Dear Dr. Entezam:

I'm pleased to inform you that your manuscript has been deemed suitable for publication in PLOS ONE. Congratulations! Your manuscript is now with our production department. 

Kind regards, 

on behalf of

Dr. Mohamad Syazwan Mohd Sanusi 

Academic Editor

PLOS ONE